# Effects of Antioxidant Combinations on the Renal Toxicity Induced Rats by Gold Nanoparticles

**DOI:** 10.3390/molecules28041879

**Published:** 2023-02-16

**Authors:** Ghedeir M. Alshammari, Mohammed S. Al-Ayed, Mohamed Anwar Abdelhalim, Laila Naif Al-Harbi, Mohammed Abdo Yahya

**Affiliations:** 1Department of Food Science & Nutrition, College of Food and Agricultural Sciences, King Saud University, Riyadh 11451, Saudi Arabia; 2Department of Physics and Astronomy, College of Science, King Saud University, Riyadh 11451, Saudi Arabia

**Keywords:** antioxidants, gold nanoparticles, lipid peroxidation, nephrotoxicity, NF-κB, NOX4, oxidative stress

## Abstract

This study investigated some possible mechanisms underlying the nephrotoxic effect of gold nanoparticles (AuNPs) in rats and compared the protective effects of selected known antioxidants—namely, melanin, quercetin (QUR), and α-lipoic acid (α-LA). Rats were divided into five treatment groups (eight rats per group): control, AuNPs (50 nm), AuNPs + melanin (100 mg/kg), AuNPs + QUR (200 mg/kg), and AuNPs + α-LA (200 mg/kg). All treatments were administered i.p., daily, for 30 days. AuNPs promoted renal glomerular and tubular damage and impaired kidney function, as indicated by the higher serum levels of creatinine (Cr), urinary flow, and urea and albumin/Cr ratio. They also induced oxidative stress by promoting mitochondrial permeability transition pore (mtPTP) opening, the expression of NOX4, increasing levels of malondialdehyde (MDA), and suppressing glutathione (GSH), superoxide dismutase (SOD), and catalase (CAT). In addition, AuNPs induced renal inflammation and apoptosis, as evidenced by the increase in the total mRNA and the cytoplasmic and nuclear levels of NF-κB, mRNA levels of Bax and caspase-3, and levels of tumor necrosis factor-α (TNF-α) and interleukin-6 (IL-6). Treatment with melanin, QUR, and α-lipoic acid (α-LA) prevented the majority of these renal damage effects of AuNPs and improved kidney structure and function, with QUR being the most powerful. In conclusion, in rats, AuNPs impair kidney function by provoking oxidative stress, inflammation, and apoptosis by suppressing antioxidants, promoting mitochondrial uncoupling, activating NF-κB, and upregulating NOX4. However, QUR remains the most powerful drug to alleviate this toxicity by reversing all of these mechanisms.

## 1. Introduction

Oxidative stress is an imbalance between reactive oxygen species (ROS) production and the cellular antioxidant system [1]. Under physiological conditions, ROS are produced within normal cells in small quantities, which are needed for cell signaling, processes, and survival. Major resources of ROS include the mitochondrial oxidative phosphorylation (OXPHOS) and other intracellular enzymes, such as nicotinamide adenine dinucleotide phosphate (NADPH) oxidase (NOX) [2,3,4,5]. However, ROS are produced in large quantities in the presence of pathological stimuli, and they can damage the cell by promoting membrane lipid peroxidation, DNA oxidation, and protein carbonylation [6]. However, this is normally faced by a strong antioxidant system composed of enzymatic and non-enzymatic components that are disrupted in various cellular compartments. Major ROS include superoxide radicals (O^2−^), hydrogen peroxide (H_2_O_2_), and hydroxyl radicals (OH^−^) [6,7]. Glutathione (GSH)—a non-protein intracellular thiol—is a major non-enzymatic antioxidant that can detoxify OH^−^ by donating electrons. However, superoxide dismutase (SOD), catalase (CAT), glutathione reductase (GRx), and glutathione peroxidase (GPx) are major enzymatic antioxidants in the cells, which collaborate to attenuate oxidative damage [8,9]. Within this view, superoxide is predominantly found in the cytoplasm and mitochondria, and it acts by converting the O^2−^ radicals to H_2_O_2_ [8]. In turn, CAT, which is resident in the peroxisomes, cytoplasm, and membranes, reduces H_2_O_2_ to water [8]. On the other hand, GPx (which can also reduce H_2_O_2_) and GRx are indispensable for regenerating GSH [7,8,9].

Chronic kidney disease (CKD) normally ends with end-stage renal disease (ESRD) and kidney failure, and it is associated with an increased mortality rate due to the development of hypertension and other forms of cardiovascular disorder [10]. Currently, the prevalence of CKD is rapidly increasing, reaching rates of 9–10% worldwide, with very high mortality rates (41.5%) [11]. Common risk factors for the development of CKD include metabolic disorders, drugs, and toxins [12]. However, the overproduction of ROS and the subsequent oxidative stress response are believed to be the major central mechanisms underlying the pathogenesis of CKD from different backgrounds by triggering cellular damage, as well as other pathological pathways, such as inflammation, fibrosis, and apoptosis [3,4,13,14,15]. In this regard, accumulating clinical and experimental data have confirmed that ROS overproduction triggered by mitochondrial dysfunction and upregulation/activation of NOX2/4 is the main mechanism underlying several renal pathological alterations and reduced function in both acute kidney disease and CKD [2,13,16,17].

Nanoparticles (NPs) are small materials (1–100 nM in size) [18]. During the last few decades, the industrial use of NPs has dramatically increased in medicine, industry, food preservation, manufacturing, and agriculture [18,19]. Several nanomaterials are currently available in the markets, including nanoplates, nanofibers, and metal nanoparticles (NPs) [18]. However, the clinical safety of these NPs is still not fully confirmed, and accumulating data have shown that they might have pro-oxidant and inflammatory systemic toxic effects [16,17,18]. Gold nanoparticles (AuNPs) are the most commonly used NPs in water purification, biological imaging, gene delivery, gene therapy, dental restorations, toothpaste, skin lubrication, food packing, cosmetics, automobiles, and beverages [3,20,21]. The cellular distribution of these NPs depends on their size. Within this view, experimental studies have shown that larger AuNPs (~50 nm) deposit mainly in the liver and Kupffer cells, whereas smaller ones can deposit in the liver, brain, heart, and kidneys [22].

Oxidative-stress-induced nephrotoxicity is the most frequently reported side effect of AuNPs. Indeed, AuNPs can promote renal and hepatic oxidative damage due to their ability to increase the production of reactive oxygen species (ROS) and scavenge endogenous antioxidants [19,22,23,24,25,26,27]. In rodents, treatment with AuNPs activated cell apoptosis and necrosis and resulted in glomerular atrophy, renal cell swelling, and tubular degeneration [22,24,28]. However, the precise mechanism underlying the pro-oxidant damaging effects of these AuNPs is still unclear. Nevertheless, alleviating oxidative stress seems to be a reasonable strategy to alleviate AuNPs-mediated nephropathy.

Today, accumulating data show a promising protective effect of natural antioxidants in alleviating various forms of renal damage and CKD [29]. α-Lipoic acid (α-LA), quercetin (QUR), and melanin are among the best-known nutritional antioxidants that have been used in experimental and clinical trials to treat several chronic oxidative-stress-related disorders [30,31]. The nephroprotective effects of all of these antioxidants have been well reported in several studies and animal models, and their precise mechanistic effects have been fully reviewed [27,32,33,34,35]. We have previously shown the hepatoprotective potential of melanin, α-LA, and QUR in AuNPs-treated animals—an effect that was attributed to the suppression of lipid peroxidation and the stimulation of enzymatic and non-enzymatic antioxidant levels [21,36]. Moreover, we have recently shown that a combination of vitamin E and α-LA could alleviate pathological renal destruction in AuNPs-treated rats by suppressing lipid peroxidation and stimulating GSH levels [37].

Following on from the previous discussion, depicting the precise mechanisms by which AuNPs promote oxidative damage to the organs, as well as finding suitable protective agents, is an interesting target. The individual or combined protective effects and the mechanisms of action of the antioxidants α-lipoic acid (α-LA), quercetin (Qur), and melanin against AuNPs-induced nephrotoxicity have not yet been demonstrated. Therefore, in this study, we aimed to examine the effects of the subchronic administration of AuNPs on renal structure and function with respect to their effects on renal mitochondrial function and NOX2/4 expression in rats. In addition, we examined whether co-treatment with α-LA, QUR, and melanin could afford a protective effect by acting on these ROS-generating pathways.

## 2. Results

### 2.1. Changes in Body Weight and Kidney Function Markers

The final body weight was not altered with any treatment as compared to control rats (Table 1). Serum albumin levels were significantly decreased, whereas serum urea and urine volume, flow, and albumin levels were significantly increased in the AuNP groups compared to the control rats (Table 1). In addition, the AuNPs-treated rats had high serum Cr, which was significantly increased in their urine samples as compared to samples obtained from the control rats (Table 1). This led to a higher albumin/Cr ratio and low Cr clearance (CrCl) levels in the AuNP groups compared to control rats. The serum levels of albumin were significantly increased, whereas serum Cr, urine volume and flow, and urinary albumin levels were significantly reduced in the AuNPs + melanin-, AuNPs + α-LA-, and AuNPs + QUR-treated rats, as compared to the AuNPs-treated rats that were given the vehicle (Table 1). The most significant improvement was seen in the latter group as compared to the other treatment groups (Table 1). Despite this, the levels of serum Cr and urea, urine volume and flow, and urinary albumin/Cr ratio remained significantly higher, but the serum levels of albumin, urinary Cr, and CrCl ratio remained significantly lower than their basal levels measured in the control rats (Table 1).

### 2.2. Changes in Renal Antioxidant and Inflammatory Markers

The levels of GSH, SOD, and CAT were significantly decreased, but the levels of MDA, TNF-α, and IL-6 were significantly increased, in the renal tissues of AuNPs-treated rats as compared to control rats (Table 2). The renal levels of GSH, SOD, and CAT were significantly increased, but the levels of MDA, TNF-α, and IL-6 were significantly and progressively decreased in the AuNPs + melanin-, AuNPs + α-LA-, and AuNPs + QUR-treated rats as compared to the AuNPs-treated rats (Table 2). The greatest increases in the GSH, SOD, and CAT levels and the greatest reductions in the renal levels of MDA, TNF-α, and IL-6 were seen in the livers of AuNPs + QUR- as compared to AuNPs + melanin- and AuNPs + α-LA-treated rats. These values remained slightly but significantly different from the basal control levels (Table 2).

### 2.3. Changes in the Transcriptional Activity of NF-κB and Apoptotic/Anti-Apoptotic Markers

The total mRNA levels of NF-κB and the total cytoplasmic and nuclear levels of NF-κB p65, along with the total mRNA levels of Bax and caspase-3, were significantly increased in the renal tissues of AuNPs-treated rats as compared with the control rats (Figure 1A–C,E,F). However, no significant changes in the mRNA levels of Bcl2 were seen between all experimental groups (Figure 1D). A significant progressive reduction in the mRNA levels of NF-κB, total cytoplasmic and nuclear levels of NF-κB p65, and mRNA levels of Bax and caspase-3 was seen in the kidneys of AuNPs + melanin-, AuNPs + α-LA-, and AuNPs + QUR-treated rats as compared to the AuNPs-treated rats (Figure 1A–C,E,F). The greatest improvements in the levels of all of these markers were seen in the kidneys of the AuNPs + QUR-treated rats as compared to their corresponding levels detected in the kidneys of AuNPs + melanin- and AuNPs + α-LA-treated rats (Figure 1A–C,E,F).

### 2.4. Changes in the Mitochondrial Coupling, Mitochondrial Membrane Potential (of mtPTP), and Expression of NOX4

The renal expression of NOX4 was significantly increased in the kidneys of AuNPs-treated rats compared to the controls (Figure 2A). Moreover, the reduction in the absorbance (V*_max_*) as an indicator of the mtPTP was significantly reduced in the renal tissues of AuNPs-treated rats compared to the controls (Figure 2B). In addition, the consumption of O_2_ during state 3 respiration was significantly increased, whereas its consumption during state 4 respiration was significantly increased in the renal tissues of AuNPs-treated rats as compared to control rats (Figure 2C,D). The expression of NOX_4_ and the levels of V*_max_* and O_2_ consumption during state 3 and 4 respiration did not vary significantly between the AuNPs- and the AuNPs + melanin-treated rats, but they were significantly reversed in both the AuNPs + α-LA- and AuNPs + QUR-treated rats (Figure 2A–D). It should be noted that the effects were more profound in the kidney tissues of AuNPs + QUR-treated rats as compared to the AuNPs + α-LA group.

### 2.5. Histological Findings

The kidneys of the control rats showed normal features, including intact glomeruli, glomerular mass, glomerular membranes, PCTs, and DCTs (Figure 3A). Shrunk and damaged glomeruli, glomerular membranes with increased glomerular space, and severe damage to the PCTs and DCTs were seen in the kidneys of AuNPs-treated rats (Figure 3B). However, significant improvements in the structure of the glomerulus and the glomerular membrane, increased glomerular mass, and increased numbers of normal PCTs and DCTs were seen in the AuNPs + melanin-, AuNPs + α-LA-, and AuNPs + QUR-treated rats (Figure 3C–E). The smallest improvement was seen with AuNPs + melanin, whereas the least damage was seen with the AuNPs + QUR-treated rats. Nonetheless, the highest scores of glomerular and tubular damage were seen in AuNPs-treated rats, and the scores were significantly reduced in all other treatment groups (Figure 3F). However, the greatest reduction in tubular damage scores was seen in the AuNPs + QUR group, followed by the AuNPs + α-LA and AuNPs + melanin groups (Figure 3F). On the other hand, the greatest improvements in the glomerular damage score were seen in both the AuNPs + QUR and AuNPs + α-LA groups, which were not significantly different when compared with one another (Figure 3F).

## 3. Discussion

This study assessed and compared the protective effects of α-LA, QUR, and melanin against AuNPs-induced nephrotoxicity in rats and examined the possible mechanisms of their action. In accordance, all of these tested drugs showed potent nephroprotective effects due to their antioxidant and anti-inflammatory effects. Within this context, our data confirmed that the nephroprotective effect afforded by melanin is due to its ROS-scavenging abilities and the upregulation of antioxidants. On the other hand, the mechanisms underlying the nephroprotective effects of α-LA and QUR include different effects, including scavenging of ROS, upregulation of the Nrf2/antioxidant axis, suppression of NOX2 NF-κB p65, and production of inflammatory cytokines. Overall, our data suggest that treatment with QUR offers the greatest nephroprotective effect among all of the tested molecules. 

The kidneys are vulnerable to small AuNPs [19]. The nephrotoxicity induced by AuNPs has been reported in numerous studies. In this context, several authors have shown that smaller AuNPs tend to have a wider distribution in the plasma and several organs, including the kidneys. At the structural and functional levels, we have shown that short-term administration of AuNPs for 7 days in male rats induces renal impairment that is associated with higher plasma Cr and urea nitrogen (BUN) levels, as well as dilation, vacuolization, and necrosis of the PCTs and DCTs [19,23,38]. Along the same lines, 30 days of treatment with AuNPs in the rats in the present study also induced similar but more severe pathological, structural, and biochemical abnormalities and impaired their kidney function [37]. They also showed increased urine flow and volume, which may have resulted from the observed kidney damage. It could also be possible that AuNPs stimulated water intake in these rats by affecting the hypothalamus and ADH release, which requires further attention. However, the new findings regarding the nephroprotective effects of α-LA, QUR, and melanin observed in this study against AuNPs-mediated nephrotoxicity provide potential therapeutic targets. This supports previous findings showing the protective effects of α-LA, QUR, and melanin against AuNPs-mediated hepatotoxicity in rats [17]. In addition, several studies have also shown a protective effect of QUR in several animal models of renal disorders or CKD [39,40,41,42]. Likewise, melanin-based nanoparticles can be used to effectively treat acute kidney injury [34].

Oxidative stress, inflammation, and apoptosis are interconnected mechanisms that lead to tissue damage in several situations, including CKD [43]. Cell apoptosis includes two major pathways: extrinsic and intrinsic cell death [44]. On the one hand, extrinsic cell death starts outside the cell when a dead ligand (e.g., TNF-α) binds to a death receptor, e.g., TNFR1) [44]. On the other hand, intrinsic cell death—also called mitochondria-mediated apoptosis—starts intracellularly when an apoptotic factor such as Bax binds to the mitochondria to trigger their opening and the release of cytochrome c. In both cases, apoptosis is achieved by the activation of special caspases (e.g., caspase-3) [44]. However, Bcl2 is the main anti-apoptotic protein in the cells that can bind and prevent the mitochondrial translocation of Bax [45]. On the other hand, NF-κB is the main inflammatory transcription factor that stimulates the synthesis of several inflammatory cytokines (e.g., TNF-α, IL-1, and IL-6). The cellular effect of NF-κB could be an apoptotic or survival (i.e., anti-apoptotic) signal, depending on several factors, including the type of stimulus and the cell type, as well as the strength of the stimulus [46,47]. In this context, NF-κB can stimulate cell survival by activating survival pathways and stimulating the expression of anti-apoptotic proteins [46,47]. On the other hand, it can promote cell death by stimulating p53, upregulating Bax, and possibly generating ROS [46,47]. Indeed, a positive crosstalk has been reported between NF-κB and ROS, where each can positively increase the production of the other, thereby creating a vicious cycle of inflammation and oxidative stress [48]. However, accumulating evidence has shown that the activation of NF-κB in the majority of renal disorders, as well as in CKD, is always a damaging effect that promotes oxidative stress, fibrosis, and apoptosis [49,50].

MDA is a lipid peroxidation marker that is produced by the peroxidation of membrane fatty acids and is reflective of the amount of ROS [7]. High levels of ROS, induced intrinsic cell death, and increased levels of MDA, C-reactive protein (CRP), IL-6, IL-1, TNF-α, and other adhesive molecules were observed in the sera and kidneys of patients and animal models with CKD, and they were positively correlated with the severity of the renal damage and disease [1,51,52,53,54,55,56,57,58]. Nonetheless, ROS have been identified as the upstream mechanism leading to CKD by depleting and overwhelming the kidneys, promoting lipid peroxidation and inflammation, and inducing intrinsic cell apoptosis [43,59]. Moreover, short-term (7 days) treatment with both small and large AuNPs induced large quantities of ROS, enhanced the levels and nuclear translocation of NF-κB, and increased levels of TNF-α and IL-6 in the livers and kidneys of treated rodents [3,14,19,20,21,22,23,38,60]. Similar results were also observed in the kidneys of rats of this study, where treatment with the small AuNPs (10 nm) for 30 days increased the levels of ROS, MDA, TNF-α, and IL-6, as well as the mRNA levels of NF-κB, and concomitantly reduced the levels of GSH, SOD, and CAT in the kidneys of the treated rats. In addition, AuNPs promoted intrinsic cell death by stimulating the expression of Bax and caspase-3, with no effect on the expression of Bcl2. This could be due to the activation of p53 through ROS-induced DNA damage, which normally upregulates Bax, thereby altering the Bax/Bcl2 ratio to induce cytochrome c release and caspase-3 activation [59]. Such apoptotic findings are the first to be reported in the kidneys of rats after subchronic administration of AuNPs. However, all of the administered treatments—including α-LA, QUR, and melanin—attenuated/reversed these effects in the AuNPs-treated rats. These data confirm the importance of these drugs in alleviating renal damage and intrinsic cell apoptosis in this animal model by reducing ROS generation, upregulating antioxidants, and suppressing NF-κB p65. However, α-LA remained the most protective agent among all of these drugs that act on these protective pathways. 

These results were expected, due to the well-reported antioxidant and anti-inflammatory renal and systemic protective effects of α-LA, QUR, and melanin [33]. In addition, QUR, α-LA, and melanin are considered to be safe and can improve the antioxidant contents in healthy kidneys, even if used at higher doses. Supporting our data, α-LA has also been shown to prevent renal, hepatic, pulmonary, cardiac, and neural damage and apoptosis in several animal models via its abilities to scavenge ROS, chelate iron, upregulate/activate the keep-1/Nrf2/antioxidants axis, suppress the NF-κB p65/inflammatory cytokine axis, downregulate Bax, upregulate Bcl2, and recycle vitamins E and C [27,61]. Likewise, QUR has potent antioxidant, anti-inflammatory, and anti-apoptotic effects that can be attributed to similar mechanisms of action to those of α-LA [29,30]. However, the antioxidant effect of melanin is not yet well established and is under investigation. Previous studies have shown that the systemic protective effects of melanin are attributable to the upregulation of antioxidants and scavenging of ROS due to the production of high levels of eumelanin—a common antioxidant [31,62]. In addition, treatment with melanin prevented adenine-induced chronic renal failure in mice and cisplatin-mediated nephrotoxicity by suppressing MAD, upregulating GSH, SO, GPx, and CAT, and inhibiting the production of TNF-α and IL-6 [63]. Hence, our data also suggest that melanin could inhibit the upregulation and the nuclear localization of NF-κB, which could be an independent effect or a secondary effect due to its antioxidant potential. Nonetheless, QUR, α-LA, and melanin also attenuated AuNPs-induced cell apoptosis by suppressing Bax and caspase-3, with no effect on the levels of Bcl2. This could be due to their direct inhibitory effects on these apoptotic markers as a result of their antioxidant and anti-inflammatory effects, which normally induce intrinsic cell death by upregulating these factors. Even though this is consistent with the findings of many other studies on the inhibitory effects of QUR and α-LA on Bax and caspase-3, our results contradict other studies that have shown these molecules’ ability to upregulate Bcl2 [64,65]. This could be explained by the variations in the animal model, treatment period, tissues, and the use doses of drugs. 

Nonetheless, it was also very interesting to investigate the exact pro-oxidant mechanism of AuNPs and the potential protective mechanisms of all of these tested drugs. Therefore, we targeted mitochondrial oxidative phosphorylation and the expression of NOX, as these are major ROS-generating pathways the mediate CKD in several disorders and toxicities [2,66]. In renal cells, the majority of ROS are produced in the mitochondria and originate from molecular oxygen (O_2_) after receiving an electron from oxidases (i.e., NADPH oxidases (NOX), monooxygenases, lipoxygenases, xanthine oxidases, etc.) and/or the electron transport chain cytochromes [61]. Superoxide radicals (O_2_) produce other oxidants, such as hydrogen peroxides (H_2_O_2_), peroxynitrite (ONOO-), hydroxyl radicals (OH^−^), and others [51]. The NOX family comprises seven members (i.e., NOX1 to NOX5, Duox1, and Duox2). However, NOX_1_, NOX_2_, and NOX_4_ are highly expressed in the kidneys and are expressed in a regional and cell-specific manner [15]. NOX4 was detected in the nephrons’ epithelial cells, mesangial cells, fibroblasts, podocytes, and endothelial cells [15]. While the functional roles of NOX1 and NOX2 remain unclear, NOX4 is the main isoform responsible for producing ROS in the renal cells, which act as a second messenger for regulating several physiological processes and signaling pathways, including oxygen sensing, gluconeogenesis, adipocyte differentiation, ion channels, glucose transport, and cell growth. However, high levels of NOX4-derived ROS and increased expression of NOX4 mediated AKD and CKD in several experimental animal models, including those of metabolic, diabetic, hypertensive, cisplatin-induced, and obstructive nephropathies [15,61]. Furthermore, they caused renal cell carcinoma by promoting oxidative damage, inflammation, fibrinogens, and apoptosis [66].

The most interesting finding in this study is that we are the first to demonstrate that AuNP intoxication causes mitochondrial damage and upregulates renal NOX4. Herein, 30 days of administration of AuNPs for the rats in this study caused a reduction in the state 3 O_2_ consumption and an increase in the time of state 4 O_2_ consumption and the maximal rate of pore opening (V*_max_*), indicating impaired respiration and loss of mtPTP. It also stimulated the transcription of NOX4, indicating an upregulation of this oxidase. Many toxicity studies of other metal NPs support these results. Indeed, silver NPs (AgNPs) also induced apoptosis in colorectal cancer cells by upregulating the expression of NOX4 [67]. Similarly, Sun et al. [68] demonstrated the ability of AgNPs to promote oxidative damage to vascular cells by upregulating NOX4. However, the progressive reduction in V*_max_* values after treatment with melanin, α-LA, and QUR is clear evidence of the ability of these molecules to attenuate the permeability of the mitochondrial membrane, which could be explained by their corresponding progressive antioxidant effects and suppression of Bax. Moreover, only α-LA and QUR—but not melanin—attenuated the impairment in the mitochondrial respiratory process and the expression of NOX, indicating the importance of these mechanisms in their antioxidant nephroprotective effects. These data suggest that the antioxidant potential of melanin is not related to the improvement of mitochondria function or regulation of NOX4 but is most likely due to the upregulation of antioxidants. 

Supporting our data, α-LA ameliorated glucocorticoid-induced renal hypertension by improving mitochondrial respiration and reducing the mitochondrial O^2−^ free radicals [69]. Similarly, α-LA improved age-associated decline in mitochondrial respiratory chain activity in the hearts of rats [70]. It also prevented salt-induced hypertension, hypertrophy, and rostral ventrolateral medulla (RVM) oxidative damage by suppressing NOX4 and the generation of ROS [71]. α-LA also prevented iron-induced acute renal injury, busulfan-induced pulmonary injury, and bleomycin-induced skin fibrosis by suppressing NOX4 [72,73]. On the other hand, QUR is a well-known antioxidant protective agent that stimulates and prevents organ damage, including nephropathy, by stimulating mitochondrial biogenesis and respiration [74,75,76]. Furthermore, QUR protected against ischemia/reperfusion-mediated cardiac and neural apoptosis and high-fat diet (HFD)-induced atherosclerosis by suppressing NOX4 [77,78,79].

## 4. Materials and Methods

### 4.1. Animals

Forty adult Wistar male rats (weighing 220–240 g; aged 12 weeks old) were supplied by the Experimental Animal Care Center at King Saud University (KSU), Riyadh, Saudi Arabia. The rats were kept in plastic cages under standard conditions (22 ± 5 °C, 55 ± 5%, and 12 h light/dark cycles). They had access to water ad libitum and were fed the usual rat diet. All experimental protocols included in this study were approved by the Research Ethics Committee at KSU (Ethics Reference No: KSU-SE-21-11), Riyadh, Saudi Arabia.

### 4.2. Experimental Design

AuNPs (10 nm) (Cat. No. MKN-Au-010) were purchased from (IPEX Corp, Canada) and were identified by electron microscopy. A total of 40 rats were included in this study and were divided into 5 groups (8 rats per group) as follows: (1) Control rats: i.p. treated with 250 µL of 5% carboxymethyl cellulose as a vehicle; (2) AuNPs-treated rats: daily treated with 250 µL of 10 nm AuNPs (3) AuNPs + α-LA-treated rats: i.p. treated with 250 µL of 10 nm AuNPs and co-treated orally with α-LA (200 mg/kg); (4) AuNPs + QUR-treated rats: i.p. treated with 250 µL of 10 nm AuNPs and co-treated orally with QUR (200 mg/kg); (5) AuNPs + melanin-treated rats: i.p. treated with 250 µL of 10 nm AuNPs and co-treated orally with melanin (100 mg/kg). All treatments were administered daily for a total period of 30 days [37]. All drugs were prepared in 5% carboxymethyl cellulose. The safety of these drugs to kidney health and their renal antioxidant stimulatory effects have been shown in several studies [80,81,82,83]. Therefore, we exempted the control treatments for data simplicity. The dose of AuNPs was selected based on our previous studies [19,23,36,37].

### 4.3. Urine, Blood, and Their Collection and Preparation

This was performed as previously conducted in our laboratories [84]. Urine samples were collected over 24 h from all rats using metabolic cages. The rats were then anesthetized with 80 mg/kg ketamine, and blood samples were collected by cardiac puncture, centrifuged at 1300× *g* for 10 min, and the supernatants were stored at −20 °C until use. Then, all rats were killed by cervical dislocation. The kidneys were collected on ice and cut into smaller pieces. Parts of the kidneys were placed freshly in 10% buffered formalin, and all other pieces were preserved at −80 °C. Later, part of the kidney tissues was homogenized in ice-cold phosphate-buffered saline (PBS, pH = 7.4) to prepare tissue homogenates. Other parts were used to extract the cytoplasmic and nuclear fractions using a commercially available kit (Cat. No. 4110147; Bio-Rad, CA, USA).

### 4.4. Biochemical Analysis of the Sera, Urine, and Homogenates

The levels of urea, creatinine (Cr), and albumin in the serum and urine samples were assessed using commercial assay kits (Cat. No. DIUR-100, BioAssay Systems, CA, USA; and Cat. No. MBS841754, MyBioSource, CA, USA, respectively). Levels of Cr excretion, as well as urine flow and Cr clearance (CrCl), were determined using equations published previously [85]. The homogenate levels of MDA, interleukin-6 (IL-6), GSH, SOD, tumor necrosis factor-alpha (TNF-α), and CAT were measured using assay kits purchased from MyBioSource, CA, USA (Cat. No. MBS2540407, Cat. No. MBS269892, Cat. No. MBS265966, Cat. No. MBS036924, Cat. No. MBS2507393, and Cat. No. MBS006963, respectively). The concentrations of NF-κB p65 in the cytoplasmic and nuclear fractions were determined using ELISA kits (Cat. No. MBS2505513 and Cat. No. MBS752046, respectively, MyBioSource, CA, USA). All analyses were performed for *n* = 8 samples/group according to each kit’s instructions.

### 4.5. Analysis of mtPTP Potential and Mitochondrial Respiration

The mitochondrial permeability transition pore potential (mtPTP potential) and the level of mitochondrial respiration, as markers of mitochondrial coupling, were measured as described by Eid et al. [86]. In brief, the mitochondrial fraction of all frozen kidneys was isolated using commercial assay kits (Cat. No. MBS508835, MyBioSource, San Diego, CA, USA). For the measurement of the mtPTP, the isolated mitochondria were suspended (1 mg/mL) in a special medium (3 mM HEPES, 215 mM mannitol, 5 mM succinate, and 71 mM sucrose) and stimulated with 75 µM tert-butyl hydroperoxide and 400 µM CaCl_2_. Then, the difference (reduction) in absorbance was measured for 10 min at 540 nm. In this test, CaCl_2_ is a trigger factor that stimulates the swelling of the mitochondria and the mtPTP opening, causing a reduction in absorbance. For the analysis of the mitochondrial coupling, state 3 and state 4 O_2_ consumptions were monitored. In accordance, the mitochondria were suspended in the respiratory buffer (5 mM K_2_HPO_4_, 1 mM EGTA, 0.2% BSA, 50 mM MOPS, 100 mM KCl, and 10 mM MgCl_2_) at 37 °C with continuous stirring. To record state 3 O_2_ consumption, O_2_ consumption was reordered after adding 2 mM pyruvate and 2 mM malate, whereas state 4 O_2_ consumption was recorded after adding 0.25 mM ADP. O_2_ consumption was recorded with the help of the PowerLab System and associated software (LabChart, version 8), along with a galvanic oxygen electrode (Model MLT1115/ST, Ad instrument, Sydney, Australia).

### 4.6. Real-Time PCR (qPCR)

Real-time PCR was performed to measure the mRNA levels of NF-κB, Bcl2, Bax, caspase-3, NOX4, and β-actin (a reference gene). Primers were provided by Thermo Fisher and are described in Table 3. RNA isolation from the frozen kidney parts was conducted using the TRIzol reagent for this part. The purity of RNA was measured at the absorbance of 260/280 nm using a NanoDrop spectrophotometer. A commercial kit was used to synthesize the first-strand cDNA (Cat. No. GE27-9261-01, Roche Diagnostic Company, Indianapolis, IN, USA). qPCR was performed using the Sofas Evergreen master mix kit (# 172-5200, Bio-Rad, Heracles, CA, USA) in a CFX69 real-time PCR machine (Bio-Rad), following the steps mentioned in the kit. In brief, the reaction mixture/well (20 µL) contained the following ingredients: 2 μL of cDNA (50 ng/well), 10 µL of the master mix reagent, 0.2 µL of the forward primer (500 nM/each), 0.2 µL of the reverse primer (500 nM/each), and 7.6 µL of nuclease-free water. The amplification steps were heating (1 cycle/98 ℃/30 s), denaturation (40 cycles/98 °C/5 s), annealing (40 cycles/60 °C/5 s), and melting (1 cycle/5 s/80–95 °C). The relative expression of each target was calculated using the 2−ΔΔCt method, and the results were presented as expressed to the reference gene (β-actin).

### 4.7. Histopathological Evaluation

Freshly collected kidneys were preserved in 10% buffered formalin for 20 h and then rehydrated in ethanol (100%, 90%, and 70%, respectively). All sections were then cut at 4–5 µm and routinely stained with hematoxylin and eosin [86]. Tubular and glomerular injury were semi-quantitatively assessed as previously described by others [87], using a scoring system from 0 to 4, where 0 = no damage to the glomerulus or tubules, 1 = mild damage (less than 25%) 2 = moderate damage (25–50%), 3 = severe damage (50–75%), and 4 = very severe damage (75–100%).

### 4.8. Statistical Analysis

GraphPad Prism analysis software (Version 8) was used for the statistical analysis of all data. The Kolmogorov–Smirnov test was utilized to test for normality. Analysis was performed using the one-way ANOVA test. The levels of significance were determined using Tukey’s post hoc test (*p* < 0.05). The results express all data as the mean ± standard deviation (SD).

## 5. Conclusions

The findings of this study are the first to show that 10 nM AuNPs can promote renal damage and dysfunction in an ROS/inflammation-dependent manner that involves the suppression of antioxidants, upregulation/activation of NF-κB p65, upregulation of NOX4, and induction of mitochondrial uncoupling and damage. However, melanin, α-LA, and QUR can provide protection against such dysfunction, and QUR remains the most effective therapy due to its ability to reverse all of these effects.

## 6. Study Limitations

Despite these data, this study still has some limitations. First, this study was conducted in male rats only, so its findings cannot confirm whether the nephrotoxic effects of AuNPs and the protection afforded by the tested drugs are gender-specific or not. Therefore, more studies on both genders, as well as at different ages, are required to validate this. In addition, our data were still observational. Hence, using transgenic knockout animals for Nrf2 and NF-κB may enrich our knowledge about the precise protective effects of these drugs in this animal model.

## Figures and Tables

**Figure 1 molecules-28-01879-f001:**
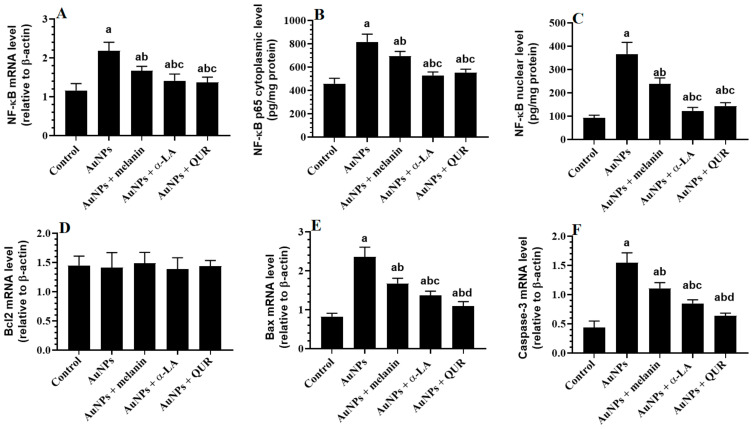
(**A**) Total mRNA levels, (**B**) cytoplasmic levels, and (**C**) nuclear levels of NF-κB as measured by ELISA, as well as total mRNA levels of (**D**) Bcl2, (**E**) Bax, and (**F**) caspase-3 in the kidneys of all groups of rats. Data are given as the mean ± SD (*n* = 8 animals/group)—^a^: vs. control rats; ^b^: vs. AuNPs-treated rats; ^c^: vs. AuNPs + melanin-treated rats; ^d^: vs. AuNPs + α-LA-treated rats.

**Figure 2 molecules-28-01879-f002:**
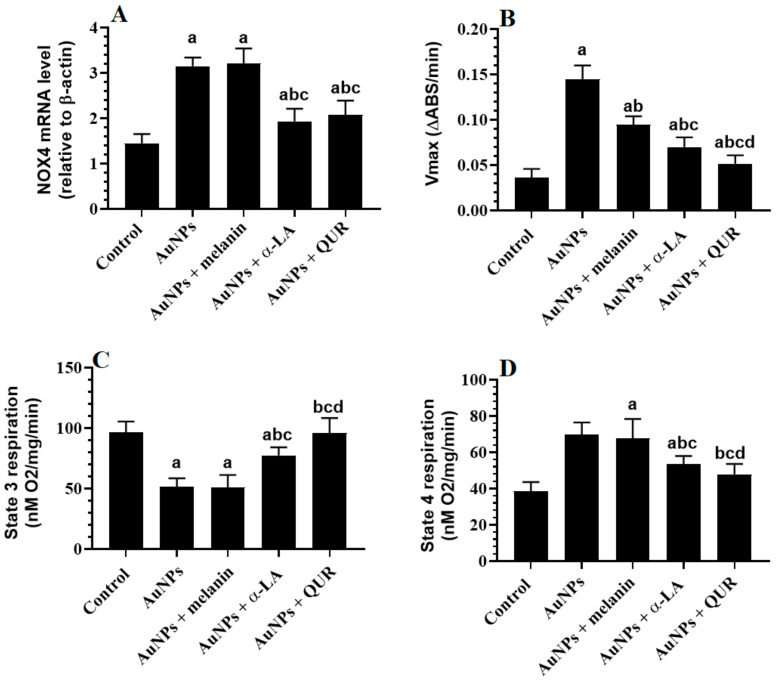
(**A**) mRNA of NOX_4_, (**B**) the reduction in VO_2_ max during the assessment of mitochondrial permeability transition pore (mtPTP) opening, and (**C**,**D**) O_2_ consumption during states 3 and 4 mitochondria respiration (mitochondrial coupling) in the kidneys of all groups of rats. Data are given as the mean ± SD (*n* = 8 animals/group)—^a^: vs. control rats; ^b^: vs. AuNPs-treated rats; ^c^: vs. AuNPs + melanin-treated rats; ^d^: vs. AuNPs + α-LA-treated rats.

**Figure 3 molecules-28-01879-f003:**
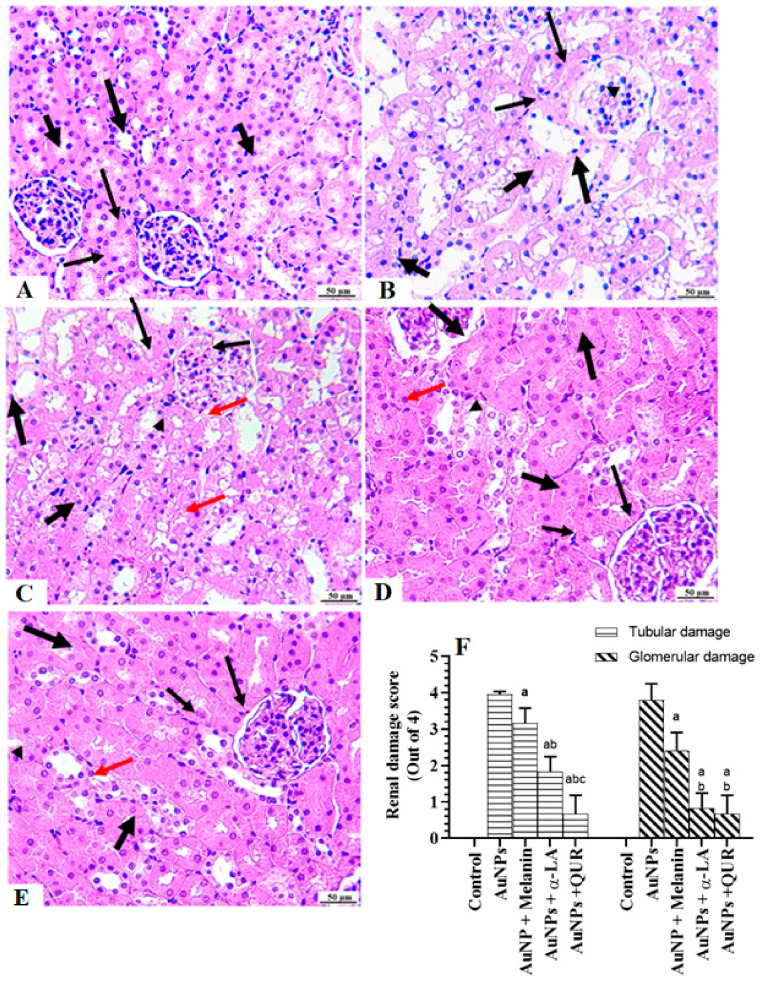
Histological sections from all groups as stained by hematoxylin and eosin: (**A**) Kidney sample taken from a control rat, showing an intact glomerulus (long thin arrow) with a normal-appearing glomerular space (short thin arrow). This group also had normal proximal convoluted tubules (PCTs) (long thick arrow) and distal convoluted tubules (DCTs) (short thick arrow). (**B**) Sample taken from an AuNPs-treated rat and showing glomerular shrinkage and damage (long thin arrow), with increased glomerular space (short thin arrow), damaged glomerular membrane (arrowhead), and severe degeneration in both the PCTs (long thick arrow) and DCTs (short thick arrow). (**C**) Sample taken from an AuNPs + melanin-treated rat and showing normal glomerulus and glomerular mass (thin arrow), as well as a normal-sized glomerular space (short, thin arrow). In addition, there was some improvement in the structure of the PCTs (short thick arrow) and DCTs (long thick arrow). Despite this, many PCTs and DCTs remained damaged (red arrows and arrowhead, respectively). (**D**,**E**) Samples taken from AuNPs + α-LA- and AuNPs + QUR-treated rats, respectively, showing normal glomeruli (long thin arrow) and glomerular spaces (short arrow), along with almost-normal PCTs (long thick arrow) and DCTs (short thick arrow). However, some damage to the PCTs and DCTs (red arrows and arrowhead, respectively) was still seen in both groups, with the least damage seen in the AuNPs + QUR group. (**F**) The glomerular and tubular damage scores in all groups of rats. In (F), data are given as the mean ± SD (*n* = 8 animals/group)—^a^: vs. control rats; ^b^: vs. AuNPs-treated rats; ^c^: vs. AuNPs + melanin-treated rats; ^d^: vs. AuNPs + α-LA-treated rats.

**Table 1 molecules-28-01879-t001:** The effects of AuNPs with or with treatment with melanin, α-lipoic acid (α-LA), and quercetin (QUR) on food intake and kidney function markers in all groups of rats.

Parameter	Control	AuNPs	AuNPs + Melanin	AuNPs + α-LA	AuNPs + QUR
Final body weight (g)	331 ± 15.6	328 ± 17.1	318 ± 21	329 ± 16	322 ± 18
Serum					
Albumin (g/dL)	6.3 ± 0.74	3.4 ± 0.48 ^a^	4.12 ± 0.37 ^ab^	5.1 ± 0.61 ^abc^	5.9 ± 0.53 ^bcd^
Urea (mg/dl)	5.44 ± 0.87	35.6 ± 3.2 ^a^	19.5 ± 2.1 ^ab^	12.8 ± 1.9 ^abc^	7.8 ± 1.1 ^abcd^
Creatinine (mg/dl)	0.58 ± 0.08	2.56 ± 0.23 ^a^	1.64 ± 0.17 ^ab^	1.1 ± 0.15 ^abc^	0.74 ± 0.09 ^abcd^
Urine					
Volume (mL/24 h)	10.4 ± 2.1	22.3 ± 2.1 ^a^	17.5 ± 1.3 ^ab^	15.1 ± 1.2 ^abc^	12.2 ± 1.5 ^abcd^
Urine flow (µL/min)	6.8 ± 0.92	15.5 ± 1.2 ^a^	11.7 ± 0.89 ^ab^	9.7 ± 1.1 ^c^	8.3 ± 0.76 ^abcd^
Albumin (Alb) (µg/dL)	10.4 ± 1.1	45.6 ± 3.7 ^a^	29.5 ± 3.1 ^ab^	18.5 ± 1.9 ^abc^	14.2 ± 1.8 ^abcd^
Creatinine (mg/dl)	74.4 ± 4.8	33.4 ± 3.3 ^a^	48.9 ± 3.4 ^ab^	55.1 ± 6.1 ^abc^	63.7 ± 5.3 ^abcd^
Urinary Alb/Cr ratio (µg/mg)	0.14 ± 0.02	1.4 ± 0.15 ^a^	0.6 ± 0.04 ^ab^	0.33 ± 0.03 ^abc^	0.22 ± 0.02 ^abcd^
CrCl (mL/min)	0.91 ± 0.18	0.22 ± 0.04 ^a^	0.38 ± 0.02 ^ab^	0.52 ± 0.06 ^abc^	0.74 ± 0.03 ^abcd^

Data are given as the mean ± SD (*n* = 8 animals/group)—^a^: vs. control rats; ^b^: vs. AuNPs-treated rats; ^c^: vs. AuNPs + melanin-treated rats; ^d^: vs. AuNPs + α-LA-treated rats.

**Table 2 molecules-28-01879-t002:** Effects of the individual or combined treatments with gold nanoparticles (AuNPs) and selected drugs on selected markers of oxidative stress and inflammation in the livers of all experimental groups.

Parameter	Control	AuNPs	AuNPs + Melanin	AuNPs + α-LA	AuNPs + QUR
MDA (µM/mg tissue)	0.51 ± 0.07	1.95 ± 0.11 ^a^	1.31 ± 0.02 ^ab^	1.04 ± 0.09 ^abc^	0.78 ± 0.05 ^abcd^
GSH (µg/mg tissue)	65.3 ± 3.9	26.3 ± 2.7 ^a^	41.6 ± 3.1 ^ab^	47.6 ± 4.9 ^abc^	56.2 ± 5.2 ^abcd^
SOD (U/mg tissue)	15.4 ± 2.1	5.1 ± 0.78 ^a^	9.4 ± 1.1 ^ab^	10.6 ± 1.1 ^abc^	12.5 ± 1.7 ^abcd^
CAT (U/mg tissue)	11.5 ± 1.6	3.7 ± 0.85 ^a^	7.6 ± 0.59 ^ab^	7.9 ± 0.52 ^ab^	9.3 ± 1.2 ^abcd^
TNF-α (pg/mg tissue)	4.5 ± 0.73	25.5 ± 3.8 ^a^	16.7 ± 1.9 ^ab^	12.1 ± 1.4 ^abc^	8.1 ± 0.94 ^abcd^
IL-6 (pg/mg tissue)	2.1 ± 0.48	13.8 ± 1.4	8.5 ± 1.2 ^ab^	5.4 ± 0.67 ^abc^	3.2 ± 0.43 ^abcd^

Data are given as the mean ± SD (*n* = 8 animals/group)—^a^: vs. control rats; ^b^: vs. AuNPs-treated rats; ^c^: vs. AuNPs + melanin-treated rats; ^d^: vs. AuNPs + α-LA-treated rats.

**Table 3 molecules-28-01879-t003:** Characteristics of the primers used for real-time PCR.

Gene	Primers (5′→3′)	Accession #	BP
NOX4	F: AGGTGTCTGCATGGTGGTG R: GAGGGTGAGTGTCTAAATTGGT	NM_053524.1	182
NF-κB	F: GAGATTGTGCCAAGAGTGAC R: CTTGTCTTCCATGGTGGATG	XM_342346.4	134
Bcl2	F: AACATCGCTCTGTGGATGAC R: GAGCAGCGTCTTCAGAGACA	U34964.1	150
Bax	F: AGGATCGAGCAGAGAGGATGG R: GACACTCGCTCAGCTTCTTGG	NM_017059	93
Caspase-3	F: AATTCAAGGGACGGGTCATG R: GCTTGTGCGCGTACAGTTTC	U49930	67
β-Actin	F: GACCTCTATGCCAACACAGT R: CACCAATCCACACAGAGTAC	NM_031144	154

## Data Availability

The datasets used and analyzed during this study are available from the corresponding author upon reasonable request.

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
