# Peer review of "Effects of Antioxidant Combinations on the Renal Toxicity Induced Rats by Gold Nanoparticles"

_molecules, 2023, doi:10.3390/molecules28041879_

Round 1

Reviewer 1 Report

The authors investigated the therapeutic efficacy of melanin, α-Lipoic acid, and quercetin against small gold nanoparticle-mediated nephrotoxicity. The study is well conducted, and the methods used are appropriate. However, the following suggestion may be useful to authors to improve the scientific content before publication.

 1.     The authors should clarify the use of Doxorubicin (DOX) use in the present study.

2.     Manuscript should carefully check for grammatical and typographical errors.

3.     Page 6, line marks 241-242. The authors may want to clarify the phrase " d: vs. α-LA+ DOX-treated rats"

4.     Page 6, line marks 241. The authors may want to change c:vs. mealnin-treated rats to c:vs. melanin-treated rats.

5.     Page 7, line marks 257. The authors may want to change c:vs. mealnin-treated rats to c:vs. melanin-treated rats

6.     Page 7, line marks 255-258. The authors may want to clarify the phrase "Table 3: Effects of the individual or combined treatments with gold nanoparticles (AuNPs) and/or doxorubicin (DOX) on selected markers of oxidative stress and inflammation in the liver of all experimental groups. Data are given as mean ± SD (n = 8 animals/group). a: vs. control rats; b: vs. AuNPs-treated rats; c:vs. mealnin-treated rats, and d: vs. α-LA+ DOX-treated rats."

7.     Page 8, line marks 274. The authors may want to clarify the phrase "d: vs. α-LA+ DOX-treated rats"

8.     Page 9, line marks 293. The authors may want to clarify the phrase "d: vs. α-LA+ DOX-treated rats"

9.     The mRNA expression of the target genes may be expressed as a relative expression level calculated using the 2−ΔΔCt method. (Ref: Livak KJ and Schmittgen TD: Analysis of relative gene expression data using real-time quantitative PCR and the 2(-Delta Delta C(T)) method. Methods. 25:402–408. 2001).

10.  Authors may comment on “The increase in nuclear levels of NF-κB in AuNPs-treated rats with an increase in mRNA levels of Bax and caspase-3”. Since, NF- κB is a pro-survival gene. Did the authors have seen apoptotic cell death in AuNPs-treated rat kidneys? 

Author Response

The authors investigated the therapeutic efficacy of melanin, α-Lipoic acid, and quercetin against small gold nanoparticle-mediated nephrotoxicity. The study is well conducted, and the methods used are appropriate. However, the following suggestion may be useful to authors to improve the scientific content before publication.

General response: Dear reviewer 1: All authors would like to thank you for you valuable comments which were all correct. We believe that these comments aimed to improve the quality of this work. For this reason, we have corrected the majority of these comments and if not, we have provided an explanation. We hope that our manuscript in the new version is acceptable for publication, and any further comments are highly appreciated.

Comment 1: The authors should clarify the use of Doxorubicin (DOX) use in the present study.

Response: Thank you for this observation, this was printed by mistake in table 3 and figure legend as we were preparing other related articles to investigate the effect of combined treatment of AUNPs and DOX on the liver and cardiac health in rats. We have corrected this all throughout the manuscript.

Comment 2: Manuscript should carefully check for grammatical and typographical errors.

Response: We have rechecked the whole manuscript for spelling, pronunciation, grammar, and other mistakes and all were corrected.

Comment 3: Page 6, line marks 241-242. The authors may want to clarify the phrase " d: vs. α-LA+ DOX-treated rats"

Response: this has been replied to in comment 1. This has been corrected

Comment 4: Page 6, line marks 241. The authors may want to change c:vs. mealnin-treated rats to c:vs. melanin-treated rats.

Response: This is, in fact, should be vs. AuNPs + melanin, this was corrected in all figure legends.

Comment 5: Page 7, line marks 257. The authors may want to change c:vs. mealnin-treated rats to c:vs. melanin-treated rats

Response: same as the previous comment. This was corrected.

Comments 6, 7, and 8.      

Page 7, line marks 255-258. The authors may want to clarify the phrase "Table 3: Effects of the individual or combined treatments with gold nanoparticles (AuNPs) and/or doxorubicin (DOX) on selected markers of oxidative stress and inflammation in the liver of all experimental groups. Data are given as mean ± SD (n = 8 animals/group). a: vs. control rats; b: vs. AuNPs-treated rats; c:vs. mealnin-treated rats, and d: vs. α-LA+ DOX-treated rats."

  1. Page 8, line marks 274. The authors may want to clarify the phrase "d: vs. α-LA+ DOX-treated rats"
  2. Page 9, line marks 293. The authors may want to clarify the phrase "d: vs. α-LA+ DOX-treated rats"

Response: All these were explained in comment 1. Again, this is printed by mistake in table 3 and figure legend as we were preparing other related articles to investigate the effect of combined treatment of AUNPs and DOX on the liver and cardiac health in rats. We have corrected this all throughout the manuscript.

Comment 9.     The mRNA expression of the target genes may be expressed as a relative expression level calculated using the 2−ΔΔCt method. (Ref: Livak KJ and Schmittgen TD: Analysis of relative gene expression data using real-time quantitative PCR and the 2(-Delta Delta C(T)) method. Methods. 25:402–408. 2001).

 Response: thank you again for this comment. We have already analyzed the data on the CFX69 bio rad PCR machine using the 2−ΔΔCt method. It was our mistake not to mention this in the method. This was corrected, and the data are presented in the right way as relatively expressed to the reference gene. Other way of presentation could be fold of change as compared to control. If the reviewer would like to have in this way, we can correct it. However, the current data presentation is commonly used.

Comment 10.  Authors may comment on “The increase in nuclear levels of NF-κB in AuNPs-treated rats with an increase in mRNA levels of Bax and caspase-3”. Since, NF- κB is a pro-survival gene. Did the authors have seen apoptotic cell death in AuNPs-treated rat kidneys? 

Response: The reviewer is correct. However, NF-κB could be a survival or apoptotic molecule, and such effect depends on numerous factors such as the nature and strength of the stimulus (please see the following articles [DOI: 10.1046/j.1432-1327.2000.01421.x   and doi.org/10.1038/cdd.2011.79). The only apoptotic evidence we had is the mRNA expression of the apoptotic markers, including Bax and caspase-3. Yes, in addition to being an inflammatory and apoptotic signal, we agree with the author that NF-κB could be also a survival signaling that compensates and resist ROS-induced apoptosis. We have introduced the role of NF-Kb and apoptosis in CKD in the discussion part. In addition, we have supported the role of NF-KB in kidney disease, where most data indicate it’s an inflammatory and apoptotic effect. This was also supported by some review articles.    

Reviewer 2 Report

This manuscript (´The melanin, αlipoic acid, and quercetin protect against gold nanoparticles-mediated nephropathy in rats`) is an interesting topic. However, the following concerns need to be addressed:

1) I suggest that the authors explain the role played by enzymatic and non-enzymatic antioxidant molecules at the cellular level in the introduction and / or discussion by explaining this in detail (e.g., how glutathione, SOD and catalase work to scavenge the toxic consequences of ROS).

2) The authors quantified MDA (an important indicator of oxidative stress). However, it would be interesting if authors also would provide data on reactive oxygen species related to the action of SOD and catalase: superoxide anion and hydrogen peroxide or at least suggested it in the discussion.

3) The suggestion provided above (topic 2) as well as the balance between the contents and activity of antioxidant enzymes are important to better understand the modulation of oxidative stress and antioxidant response. Was the activity of antioxidant enzymes, in addition to antioxidant enzyme levels, quantified?

4) Could the authors please explain why only two antioxidant enzymes (superoxide dismutase and catalase) and no other antioxidant enzymes were considered in this study? For example, the authors assessed glutathione levels. They might also assess the amount and activity of glutathione-related enzymatic antioxidants or at least suggested it in the discussion.

5) Regarding biochemical analysis, the QA/QC procedures used and the operating parameters of analytical instrumentation must be addressed within the materials section. The latest measurement techniques using chemical-specific separation and detection are also expected to be addressed in the methodology section.

Author Response

This manuscript (´Effect of antioxidant combination on the renal toxicity induced rats by gold nanoparticles`) is an interesting topic. However, the following concerns need to be addressed:

General response: Dear reviewer 1: All authors would like to thank you for your valuable comments, which were all correct. We believe that these comments aimed to improve the quality of this work. For this reason, we have corrected the majority of these comments, and if not, we have provided an explanation. We hope that our manuscript in the new version is acceptable for publication, and any further comments are highly appreciated

Comment 1:  I suggest that the authors explain the role played by enzymatic and non-enzymatic antioxidant molecules at the cellular level in the introduction and / or discussion by explaining this in detail (e.g., how glutathione, SOD and catalase work to scavenge the toxic consequences of ROS).

Response: Thank you for this comment. Yes, all authors agreed on this. We have introduced oxidative stress and the role of all these antioxidants in the introduction part to give the reader an idea about the scope of the study, In addition, we have modified the discussion to follow any further explanations.  

Comment 2:  The authors quantified MDA (an important indicator of oxidative stress). However, it would be interesting if authors also would provide data on reactive oxygen species related to the action of SOD and catalase: superoxide anion and hydrogen peroxide or at least suggested it in the discussion.

Response: we have also introduced this in the introduction part and discussed it briefly in the discussion.

Comment 3:  The suggestion provided above (topic 2) as well as the balance between the contents and activity of antioxidant enzymes are important to better understand the modulation of oxidative stress and antioxidant response. Was the activity of antioxidant enzymes, in addition to antioxidant enzyme levels, quantified?

Response: As our data indicate, we have only measured antioxidant levels not activities.

Comment 4) Could the authors please explain why only two antioxidant enzymes (superoxide dismutase and catalase) and no other antioxidant enzymes were considered in this study? For example, the authors assessed glutathione levels. They might also assess the amount and activity of glutathione-related enzymatic antioxidants or at least suggested it in the discussion.

Response: the reviewer is correct. According to the fund we have for this project, we could only measure the levels of GSH, SOD, and CAT. We will continue our research in the next stage once we get extra funds. Our aim is to completely examine the precise antioxidant mechanisms, including Nrf2/antioxidant axis, the GSH system, and the thioredoxin system. At this stage, we have provided evidence for these drugs' role in ameliorating GSH levels and SOD, and CAT enzymes.

Comment 5) Regarding biochemical analysis, the QA/QC procedures used and the operating parameters of analytical instrumentation must be addressed within the materials section. The latest measurement techniques using chemical-specific separation and detection are also expected to be addressed in the methodology section.

Response: Did the reviewer means that we should describe the methodology of each kit used for the biochemical analysis of the selected parameters. If yes, we have just indicated the assay and ELISA kits used in this procedure. We can’t write the protocol of each kit. We have provided a cat No. Where the reader can refer to it to read the protocol further. This is acceptable in most journals. In addition, we have word limits where we can’t write the full procedure.

Reviewer 3 Report

Chronic kidney disease is one of the most dreadful diseases worldwide. Using antioxidant supplementation is interesting in improving renal function. However, the current manuscript requires major corrections and language modifications.   

Comment 1: - Author used control and four different treated groups. However, what is the solo effect of melanin, quercetin, and a-lipoic acid on the kidney? Provided reference (35) also used these compounds in combination with gold nanoparticles but did not mention the effect of the individual compound on the kidney. 

Comment 2: - Present study is conducted on Wistar male rats. Do these effects are gender specific.  

Comment 3: - Histopathological evaluation needs to be corrected in the materials and methods section. In the process of wax block preparation, tissue needs to be dehydrated (not rehydrated) to remove the water from tissue (50-100% alco) and clear in a clearing agent (like xylene) afterward embedded in wax. For stain again, tissue first needs to rehydrate, and after H&E staining again, dehydrate and mount. 

Comment 4: - Group leveling needs to be corrected. In some places, the author used control and someplace control 1. 

Comment 5: - In group distribution author mentioned they used n=8/ group. However, in figure (1 & 2) legend they are mentioning n=10/group

Comment 6: - In qPCR results, authors need to mention its total mRNA levels and, if they measure cytoplasmic and nuclear individually, how they measure nuclear transcript.

Comment 7: - In table 2, the groups have no difference in weights. However, urine volume (ml/24h) increased more than double in the AuNPs group. Do AuNPs make animals drink more compared to control 

Comment 8: - ROS increases the Bax/Bcl2 ratio in mitochondria that activates caspase-3, resulting in apoptosis. Authors claimed that AuNPs promoted intrinsic cell death by stimulating the expression of Bax and caspase-3 with no effect on the Bcl2 expression. However, previously published literature shows that a-lipoic acid and quercetin significantly increase Bcl2 levels. 

Author Response

Reviewer 3

I read interestingly the manuscript entitled " Effect of antioxidant combination on the renal toxicity induced rats by gold nanoparticles". The study seems conducted carefully.

General response: Dear reviewer 1: All authors would like to thank you for your valuable comments, which were all correct. We believe that these comments aimed to improve the quality of this work. For this reason, we have corrected the majority of these comments, and if not, we have provided an explanation. We hope that our manuscript in the new version is acceptable for publication, and any further comments are highly appreciated

Comment 1) Abstract: this needs to be more informative.

Response: we have revised the abstract. According to the word limit, we have reformatted the abstract as suggested to be more informative within the allowed word limit and to fully describe the major results and findings of this study

Comment 2) Keywords: keywords need to rearrange according to the English alphabet. Also, each keyword starts with capital letters

Response: This was corrected.

Comment 3) It is suggested to rewrite the Methods section in shorter and mentioned the related references.

Response: We agree with the authors on this. However, based on the review of other reviewers, we have kept this as its. This is within the allowable world limits of the journal. If the reviewer would like, we then reconsider it in a further review if agreed with other reviewers

Comment 4) The information some primers have a serious problem (Bcl2, Bax, Β-actin).

Response: we could not understand this point. Could the reviewer clarify further? All primers were checked and have been used by others as cited in the references. Please clarify further.

Comment 5) It is suggested that the damages of different kidney parts be changed from qualitative to quantitative and compared with statistical analysis.

Response: we have added a new figure for the assessment of morphological abnormalities (glomerular and tubular damage) in the figure, and the protocol and the reference were added in the histology part. 

Comment 6) The paper has been carefully revised to improve the punctuation errors, grammar, and readability.

Response: we have checked the manuscript again, and all mistakes in the spelling, pronunciation, and typos were corrected.

Reviewer 4 Report

I read interestingly the manuscript entitled " The melanin, αlipoic acid, and quercetin protect against gold nanoparticles-mediated nephropathy in rats ". The study seems conducted carefully.

Abstract: this needs to be more informative.

Keywords: keywords need to rearrange according to the English alphabet. Also, each keyword starts with capital letters

Methods:

-It is suggested to rewrite the Methods section in shorter and mentioned the related references.

-The information some primers have a serious problem (Bcl2, Bax, Β-actin)

-It is suggested that the damages of different kidney parts be changed from qualitative to quantitative and compared with statistical analysis.

-The paper has been carefully revised to improve the punctuation errors, grammar, and readability.

Author Response

(The authors gave the same response as above.)

Round 2

Reviewer 2 Report

The authors have considered all my comments. I can see the substantial improvements that have been done in the manuscript. I have no more comments and recommend acceptance.

Author Response

The authors have considered all my comments. I can see the substantial improvements that have been done in the manuscript. I have no more comments and recommend acceptance.

Response: Thank you for accepting this work for publication.

Reviewer 3 Report

Materials and Methods still need to improve 

Author Response

Comment: Materials and Methods still need to improve

Response: Absolutely right, we have reduced the size of the materials and methods and added supporting references.

Reviewer 4 Report

Dear authors

The manuscript is interesting but the according to NCBI databases the characteristics in some of the primers have serious problems.

Author Response

Comment: The manuscript is interesting but the according to NCBI databases, the characteristics in some of the primers have serious problems.

Response: thank you for this comment, we have revised the primer sequence provided by us. Yes, you are correct, we have the wrong entry for 2 primers which were corrected.
